# Synthekines are surrogate cytokine and growth factor agonists that compel signaling through non-natural receptor dimers

Ignacio Moraga[1,2,3†], Jamie B Spangler[1,2,3†], Juan L Mendoza[1,2,3†], Milica Gakovic[1,2,3], Tom S Wehrman[4], Peter Krutzik[4], K Christopher Garcia[1,2,3*]

[1]Howard Hughes Medical Institute, Stanford University School of Medicine, Stanford, United States; [2]Department of Molecular and Cellular Physiology, Stanford University School of Medicine, Stanford, United States; [3]Department of Structural Biology, Stanford University School of Medicine, Stanford, United States; [4]Primity Bio, Santa Clara, United States

**Abstract** Cytokine and growth-factor ligands typically signal through homo- or hetero-dimeric cell surface receptors via Janus Kinase (JAK/TYK), or Receptor Tyrosine Kinase (RTK)-mediated trans-phosphorylation. However, the number of receptor dimer pairings occurring in nature is limited to those driven by natural ligands encoded within our genome. We have engineered synthethic cytokines (synthekines) that drive formation of cytokine receptor dimer pairings that are not formed by endogenous cytokines and that are not found in nature, and which activate distinct signaling programs. We show that a wide range of non-natural cytokine receptor hetero-dimers are competent to elicit a signaling output. We engineered synthekine ligands that assembled IL-2R$\beta$/IL-4R$\alpha$ or IL-4R$\alpha$/IFNAR2 receptor heterodimers, that do not occur naturally, triggering signaling and functional responses distinct from those activated by the endogenous cytokines IL-2, IL-4, and IFN. Furthermore, hybrid synthekine ligands that dimerized a JAK/STAT cytokine receptor with a receptor tyrosine kinase (RTK) also elicited a signaling response. Synthekines represent a new family of synthetic ligands with pre-defined receptors, but 'orphan' functions, that enable the full combinatorial scope of dimeric signaling receptors encoded within the human genome to be exploited for basic research and drug discovery.

*For correspondence: kcgarcia@stanford.edu

†These authors contributed equally to this work

**Competing interests:** The authors declare that no competing interests exist.

## Introduction

Cytokines and growth factor agonists activate JAK/STAT, or RTK linked homo- or hetero-dimerize cell surface receptors, respectively, to elicit signaling through intracellular trans-phosphorylation by receptor-associated kinases. The identity of the specific receptor chains within the dimer determines the signaling and functional response (*Moraga et al., 2014*; *Stroud and Wells, 2004*; *Ullrich and Schlessinger, 1990*; *Wang et al., 2009*; *Wells and de Vos, 1993*). In the case of cytokines, they act as bi-specific ligands, and determine which receptors are included in the dimers by binding to each of the two receptor extracellular domains, thus acting to bridge or cross-link the dimeric signaling complex. Cytokine receptor dimerization leads to the activation of an intracellular JAK/STAT signaling pathway, comprised of four Janus Kinases (JAK1-3, TYK2) (*Ihle et al., 1995*; *O'Shea and Plenge, 2012*) and seven signal transducer and activator of transcription (STAT1-6) proteins (*Delgoffe et al., 2011*; *Levy and Darnell, 2002*; *Murray, 2007*). While cytokines are specific for the extracellular domains of their receptors, the JAK/TYK/STAT signaling modules are found in many combinations in

endogenous cytokine receptor signaling complexes, and thus are capable of extensive cross-talk. Ligands for Receptor Tyrosine Kinase (RTK) receptors (such as EGF, VEGF, etc.) also initiate signaling through receptor dimerization, although the molecular mechanisms can be quite distinct from cytokines. In the cases of both JAK/STAT receptors and RTKs, the role of their respective ligands is to induce a positioning of the receptor subunits into a dimeric orientation and proximity that enables trans-phosphorylation of both the kinases and the receptor intracellular domains (*Lemmon and Schlessinger, 2010*). The sequence requirements (i.e. substrate specificity) of these Tyrosine kinases can be degenerate to different degrees (*Songyang and Cantley, 1995*). This offers the possibility that receptor-associated JAK/TYK and RTK kinases can be redirected, via engineered extracellular receptor dimerizing ligands, to phosphorylate alternative receptor targets than those normally driven by the endogenous ligands. Underscoring this possibility, several examples of cross-talk between JAK/STAT and RTK family receptors have been shown to occur naturally (*Vignais et al., 1996*; *Grant et al., 2002*; *Jahn et al., 2007*; *Wang et al., 2013*).

Given that the ligands determine the composition of the receptor subunits comprising the signaling dimers, and the intracellular JAK/TYK and RTK enzymes are degenerate, the number of cytokine and growth factor receptor dimer pairings that occur in nature only represents a small proportion of the total number of signaling-competent receptor pairings theoretically allowed by the system. For example, the human genome encodes for approximately forty different JAK/STAT cytokine receptors. In principle, approximately 1600 unique homo- and hetero-dimeric cytokine receptor pairs could be generated with the potential to signal through different JAK/TYK/STAT combinations (*Bazan, 1990*, *1993*; *Huising et al., 2006*). However, the human genome encodes for less than fifty different cytokine ligands (*Bazan, 1990*, *1993*; *Huising et al., 2006*), limiting the scope of cytokine receptor dimers to those that can be assembled by the natural ligands. A similar argument can be made for the RTK family of receptors and ligands.

For JAK/STAT cytokine receptors, it has been previously established that genetically modified chimeric receptors in which the extracellular domain (ECD) of a cytokine receptor has been fused with the intracellular domain (ICD) of a different receptor activated signaling in a ligand-dependent manner (*Fujiwara et al., 1997*; *Heller et al., 2012*; *Kawahara et al., 2006*; *Ohashi et al., 1994*; *Pattyn et al., 1999*; *Schaeffer et al., 2001*). However, for this concept to be practically useful, soluble ligands that co-opt endogenous receptors and assemble non-natural dimers on natural cells and tissues are required. This could be accomplished by **synthe**tic cyto**kines**, or *synthekines,* that drive formation of artificial cytokine receptor pairs not formed by natural endogenous cytokines. Synthekines could potentially activate new signaling programs and elicit unique immunomodulatory activities compared to genome-encoded cytokines, providing an almost unlimited supply of ligands to expand the functional scope of cytokine action.

Previous studies have reported the engineering of cytokine variants termed 'fusokines' that promote new activities by genetically fusing two intact cytokines via a polypeptide linker, resulting in dimers of natural receptor signaling dimers (*Ng and Galipeau, 2015*; *Rafei et al., 2009*). In this approach the two connected cytokines each retain activity, so the observed activities of the fused cytokine are due to both the independent and additive combinations of the two natural cytokine receptor signaling dimers. The use of engineered ligands (synthekines) that dimerize two different cytokine receptors in a molecularly defined 1:1 stoichiometry on the surface of responsive cells presents an alternative approach that leads to unique, rather than additive, signaling outputs. Formation of a defined dimeric complex, like the one formed by genome-encoded cytokines, allows precise mechanistic insight into the nature of the signaling complex eliciting the new signaling programs and activities engaged by these ligands.

To explore the generality of this idea, we expressed and characterized a 10 × 10 matrix of chimeric cytokine receptor pairs using an orthogonal extracellular domain as a common hetero-dimerizing unit, fused to cytokine receptor intracellular domains, allowing for the evaluation of the signaling of 100 different cytokine receptor dimers. Most cytokine-receptor pairs sampled in this chimeric receptor matrix activated signaling. In a second step, we genetically fused two antagonist versions of IL-2, IL-4, and interferon (IFN), that form 1:1 receptor-ligand complexes, with a polypeptide linker to engineer synthekines that dimerize non-natural cytokine receptor pairs on the cell surface. Stimulation of cell lines and peripheral blood mononuclear cells (PBMCs) with engineered synthekines revealed signaling and cellular signatures with features both shared and distinct from the parent cytokines. We extended the synthekine concept to dimerize c-Kit, a tyrosine kinase receptor for Stem Cell Factor

(SCF), and thrombopoietin receptor (TpoR), a JAK/STAT cytokine receptor, which resulted in a measurable signaling output that qualitatively differed from that induced by their respective endogenous ligands. Our results serve as proof of concept that dimerization of non-natural JAK/TYK/STAT and RTK receptor pairs is a viable approach to generate new signaling programs and activities whose functional consequences can be explored as if they are newly discovered orphan cytokine ligands, but with pre-defined receptors.

## Results

### Signal activation induced by chimeric cytokine receptors

We first wished to determine the potential for JAK/STAT cross-talk between a large number of enforced non-natural cytokine receptor dimers (*Figure 1A*). We generated an array of chimeric receptors, in which the extracellular domains (ECDs) of the IL-1 receptors (IL-1R1 and IL-1R1Acp) were fused to the transmembrane (TM) and intracellular domains (ICDs) of ten different cytokine receptors, generating a 10 × 10 matrix of possible receptor pair combinations (*Figure 1B*). We used the IL-1 system for two reasons: (1) IL-1 binds with very high affinity to its receptors, which allows for signal activation even at low receptor expression levels; and (2) IL-1 does not signal via the canonical JAK/STAT pathway, eliminating background activity resulting from dimerization of endogenous IL-1 receptors.

Jurkat cells, which express all JAKs and STATs except STAT4, were electroporated with the indicated combinations of chimeric receptors and analyzed for IL-1R1 and IL-1R1Acp surface expression by flow cytometry (*Figure 2—figure supplement 1*) and for IL-1-dependent signal activation by Western blot (*Figure 2A* and *Figure 2—figure supplement 2*). Although most IL-1 receptor combinations exhibited robust cell surface expression, very low levels of expression were detected for some receptor pairs, although signaling by IL-1 stimulation remained detectable (*Figure 2—figure supplements 1–2*). Although different receptor combinations activated signaling to different degrees, binary heat maps depicting phosphorylation of six STATs are presented in *Figure 2A*. Red squares indicate signaling, black squares indicate no signaling. As expected, STAT2 and STAT4 were not activated by any receptor combination due to low STAT2 expression and lack of STAT4 expression in Jurkat cells (*Figure 2A* and [*Marijanovic et al., 2007*]). STAT1 and STAT6 proteins were activated only by chimeric receptor pairs containing IFNAR2 and IL-4Rα respectively, consistent with the specific activation of these two STATs by IFNs and the IL-4 and IL-13 cytokines (*Figure 2A*). In contrast, STAT3 and STAT5 proteins were activated by many chimeric receptor pair combinations, consistent with the more pleiotropic use of these two STATs by cytokines (*Figure 2A*). Although the majority of receptor pair combinations activated signaling, we also found receptor pairs that did not induce productive signaling despite robust surface expression (*Figure 2—figure supplement 1*), such as IL-2Rβ homodimers and IL-13Rα1 homodimers (*Figure 2A* and *Figure 2—figure supplement 2*). Overall our data show that most receptor dimer combinations tested activated STAT proteins, revealing the high plasticity of the cytokine-cytokine receptor system.

### Signal activation by JAK2/JAK3 cytokine receptor pairs

One striking observation from our chimeric receptor study was that chimeric receptor combinations pairing JAK2 and JAK3, (*i.e.* erythropoietin receptor (EpoR)/γc and IL-23R/γc), did not appear to activate signaling. Interestingly, the JAK2/JAK3 pairing is not found in nature, raising the question of whether lack of signal activation by this pair could result from structural incompatibilities between these two kinase molecules that would prevent cross-activation. To test this hypothesis we inserted alanine residues in the juxtamembrane domain of EpoR, which has been shown to modulate signaling by altering the register of the juxtamembrane region of the receptor (*Constantinescu et al., 2001*). Specifically, between one and four alanines were inserted after $R^{251}$ on the EpoR ICD (*Figure 2B*). These EpoR mutants were fused to the IL-1R1 ECD and co-transfected with either IL-1R1Acp-IFNAR2 (positive control) or IL-1R1Acp-γc in Jurkat cells (*Figure 2—figure supplement 3*). Insertion of one, three or four alanines in the juxtamembrane domain of EpoR did not affect its ability to signal when paired with IFNAR2, but insertion of two alanines, prevented signaling by this receptor pair (*Figure 2C*). Insertion of one or three alanines in the juxtamembrane domain of EpoR did not recover signaling by the EpoR/γc (JAK2/JAK3) receptor pair, insertion of four alanines

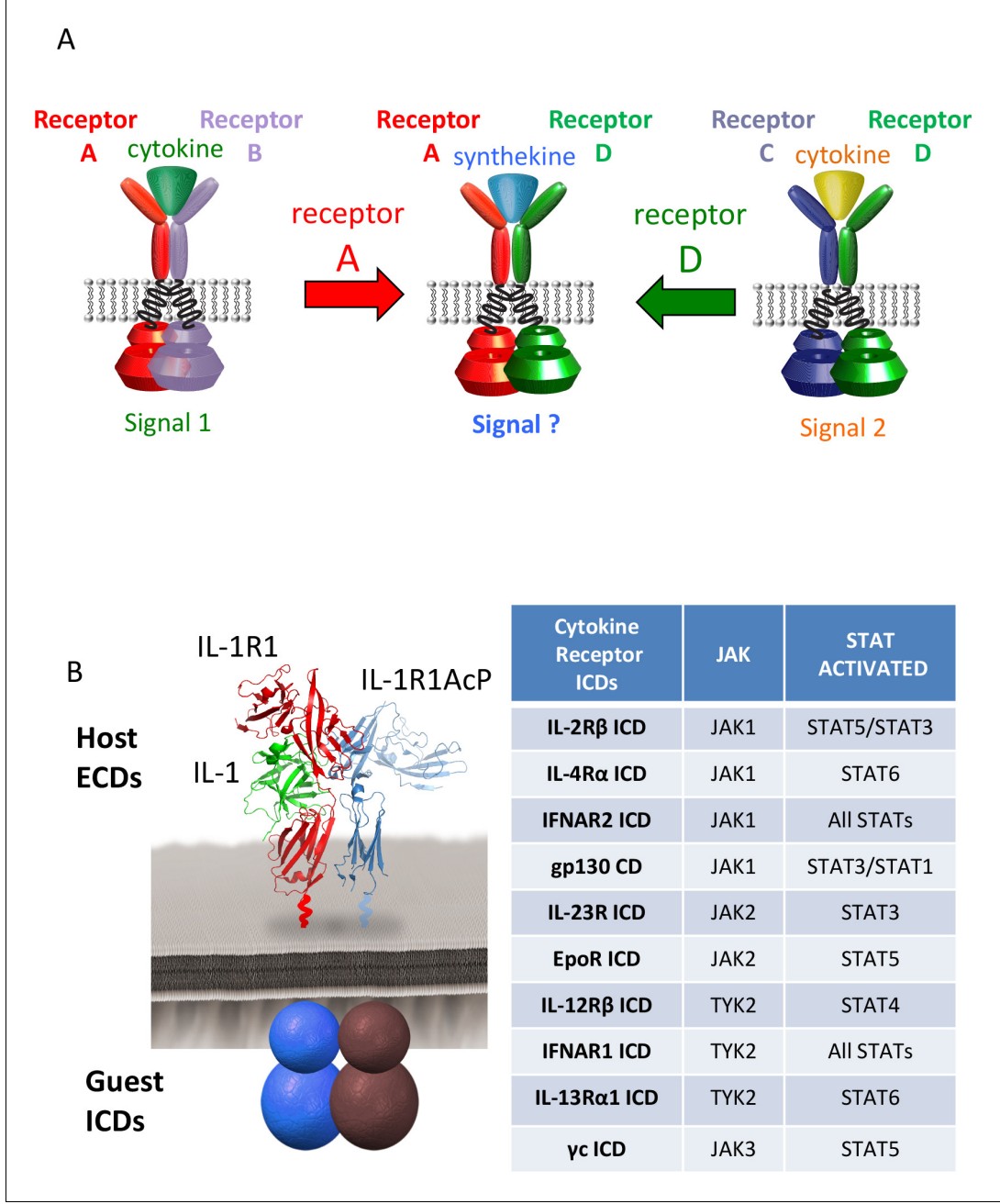

**Figure 1.** Dimerization of non-natural receptor pairs by engineered surrogate ligands. (**A**) Schematic detailing the dimerization of new cytokine receptor pairs by synthekines. A hypothetical synthekine recruits receptors A and D to form a new ternary complex distinct from that formed by each of the cytokines. (**B**) Schematic representation of the IL-1-mediated complexation of IL-1R1 and IL-1R1AcP chimeric receptors. The intracellular domains of the cytokine receptors indicated in the right table were grafted onto the IL-1R1 or IL-1R1AcP extracellular domains. JAKs and STATs activated by each receptor are indicated in the table.

marginally recovered signaling, and insertion of two alanines fully recovered signal activity by this receptor pair (*Figure 2C*). These results suggest the existence of structural constraints between JAK2 and JAK3 limit these two kinases from triggering signaling in our chimeric receptor system. However, this experiment demonstrates that varying the ligand-receptor geometry in non-natural receptors pairs using JAK2 and JAK3 is a viable option to recover signaling. Previously we have

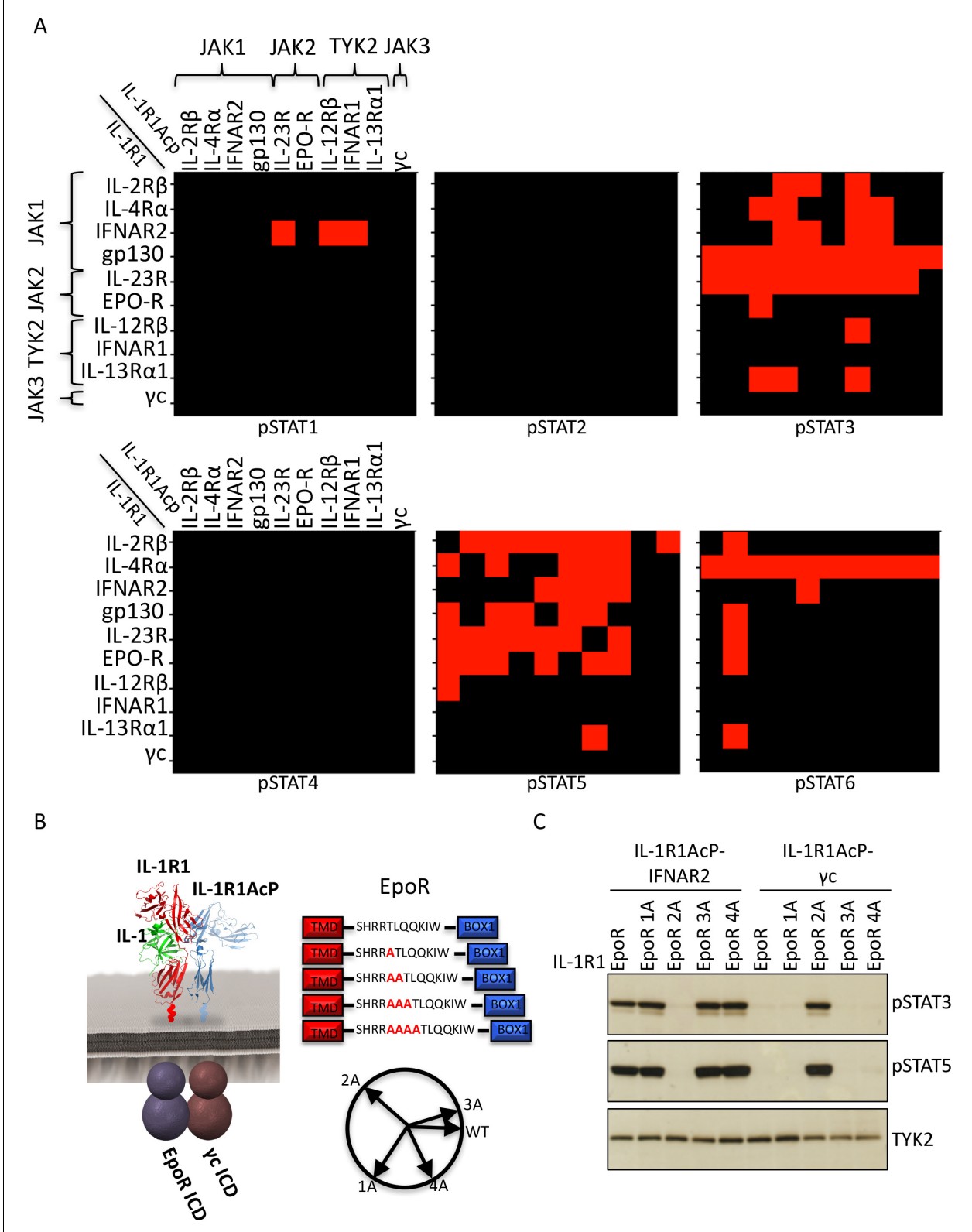

**Figure 2.** Non-natural cytokine receptor pairs activate signaling. (**A**) Heatmap representation of STAT molecules activated by the 100 different cytokine receptor pair combinations generated from the chimeric receptor matrix described in *Figure 1B*. Results were binary coded to 1, presence of band, or 0, absence of band, in western blot analysis. (**B**) Schematic representation of the designed IL-1-inducible chimeric receptors (left). Alanine insertion mutagenesis of the EpoR juxtamembrane domain is detailed in the center. Alanine residues (1A, 2A, 3A, or 4A) were inserted after R[251]. Alpha-helical

*Figure 2 continued on next page*

Figure 2 continued

wheel projections of the register twists introduced by alanine residue addition are presented at right bottom. Each residue adds a 109° rotation, with insertion of 3A residues bringing the register close to the original position. (C) Phospho-STAT3 (pSTAT3) and pSTAT5 levels measured by western blot in IL-1-activated Jurkat cells expressing the indicated chimeric receptor pairs. Insertion of two alanines recovers signaling by the IL-1R1-EpoR/IL-1R1AcP-γc receptor pair. Total levels of TYK2 are presented as a loading control. The western blot presented is a representative example of two independent experiments.

The following figure supplements are available for figure 2:

**Figure supplement 1.** Cell surface expression of chimeric receptors in Jurkat cells.

**Figure supplement 2.** Signaling profiles activated by chimeric receptors in Jurkat cells.

**Figure supplement 3.** Cell surface expression of EpoR chimeric receptors in Jurkat cells.

**Figure supplement 4.** Alanine insertions do not recover signaling by the IL-23R-IL-12Rβ, IL-2Rβ-IL2Rβ and EpoR-EpoR chimeric receptors.

shown that altering the dimer geometry of EpoR with surrogate ligands results in differential signaling outputs, so synthekines could also exploit this parameter.

In addition to JAK2/JAK3 pairs, we observed other chimeric receptor pairs that did not appear to activate signaling. We asked whether insertion of alanines in the juxtamembrane domain would recover signaling by these receptors as well. Insertion of alanines in the juxtamembrane domains of IL-2Rβ, IL-12Rβ, and EpoR did not recover signaling by the IL-2Rβ-IL-2Rβ, IL-12Rβ-IL-23R and EpoR-EpoR pairs (*Figure 2—figure supplement 4*). Strikingly, the IL-12Rβ-IL-23R and EpoR-EpoR pairs represent the receptor dimers engaged by IL-23 and EPO respectively. We think it is most likely that the IL-1 receptor extracellular orientation and proximity is not favorable for some natural and non-natural cytokine receptor pairs. This is a technical limitation of the chimeric receptor strategy we used and we think that the lack of signaling for some of the pairs is not due to intrinsic inability for particular JAK/TYK/STAT combinations to function, however this remains to be experimentally tested. The collective results from the chimeric IL-1 receptor experiments is that many, if not most, non-natural cytokine receptor pairs can signal through one or more STATs, but that certain pairs will have distinct dimer orientation and proximity requirements that will depend on the synthekine.

## Signal activation profiles induced by synthekines

We wished to explore whether dimerization of non-natural cytokine receptor pairs by synthekines would activate signaling in unmodified cells (*Figure 3A*). We used a bi-specific strategy where we fused two cytokines together, each of which could only bind to one of its two receptors, thus creating a defined 1:1 receptor dimer. To implement this approach, we engineered antagonist, or 'dominant negative (DN)' versions of IL-4, IL-2, and IFN that preserve binding to their high affinity receptor subunits (IL-4Rα for IL-4, IL-2Rβ for IL-2, and IFNAR2 for IFN) but for which binding to their low affinity receptor subunits has been disrupted (IL-13Rα1 and γc for IL-4, γc for IL-2, and IFNAR1 for IFN). These 'DN' cytokines function as high-affinity binding modules devoid of signaling activity on their own. As anticipated, the wild type cytokines activated signaling in the Hut78 T cell line (*Figure 3B*), but the dominant negative mutants were unable to promote signal activation, or at least to nearly undetectable levels (*Figure 3C*). For the IL-2 cytokine, we used an engineered variant denoted Super-2 that has 200-fold enhanced affinity for the IL-2Rβ receptor subunit (*Levin et al., 2012*) (*Figure 3B*). We then genetically fused pairs of dominant negative cytokine mutants with a Gly$_4$/Ser linker to generate new ligands that induce formation of IL-2Rβ-IL-4Rα and IL-4Rα-IFNAR2 non-natural receptor dimers (*Figure 3A*).

When added to unmodified cell lines, these bi-specific-DN synthekines activated signaling profiles that were qualitatively and quantitatively distinct from those induced by the parent cytokines (*Figure 3D*). Stimulation of Hut78 cells with SY1 SL, which promotes dimerization of IFNAR2 and IL-4Rα, resulted in STAT5 and STAT6 activation (*Figure 3D*). Notably, lengthening the polypeptide linker connecting these two dominant negative proteins by 10 residues (SY1 LL) increased the signaling potency exhibited by this synthekine (*Figure 3D*). Stimulation of Hut78 cells with SY2 synthekine,

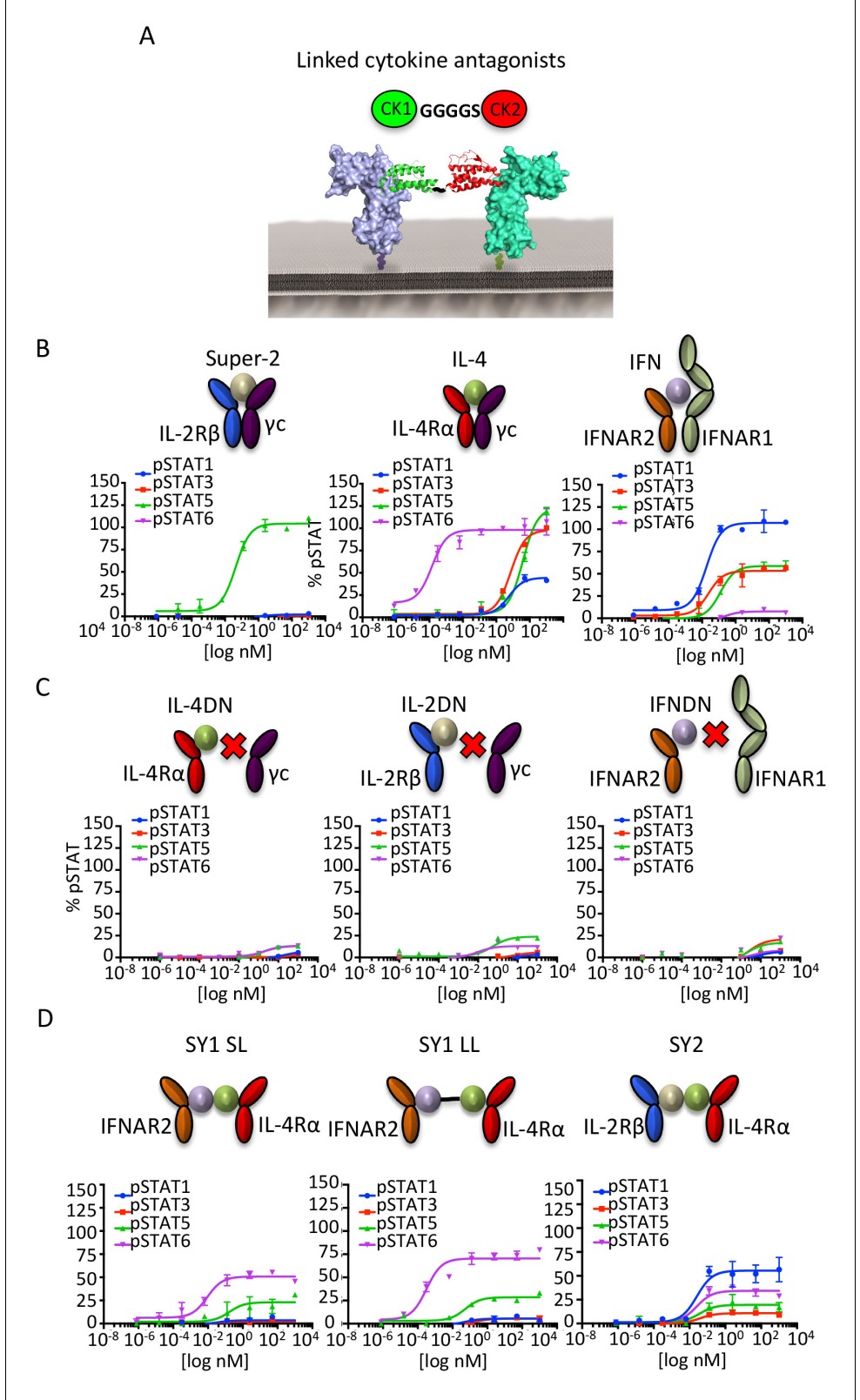

**Figure 3.** Synthekines dimerizing non-natural cytokine receptor pairs activate signaling. (**A**) Layout and complex formation by a synthekine. Two dominant negative cytokine variants are genetically fused by a Gly$_4$/Ser linker, resulting in a new molecule that induces formation of a non-natural cytokine receptor heterodimer. (**B**)-(**D**) pSTAT1, pSTAT3, pSTAT5 and pSTAT6 levels activated by the IL-4, Super-2 (affinity-matured variant of IL-2), and IFNω

*Figure 3 continued on next page*

*Figure 3 continued*
cytokines (**B**), the dominant negative cytokine variants IL-4DN, IL-2DN, and IFNDN (**C**), or the SY1 SL, SY1 LL, and SY2 synthekines (**D**) in the Hut78 T cell, as measured by flow cytometry. Data (mean ± SD) are from two independent replicates.
The following figure supplement is available for figure 3:

**Figure supplement 1.** Signaling profiles activated by stimulation with Super-2/IL-4 and IL-4/IFN cytokine combinations.

which dimerizes IL-2R$\beta$ and IL-4R$\alpha$, resulted in STAT1 activation and weaker STAT3, STAT5 and STAT6 activation (*Figure 3D*). In all instances, the synthekines elicited maximum responses (Emax) significantly lower than those activated by genome-encoded cytokines. The different signaling programs activated by the synthekines do not appear to be the result of additive effects from the two parental cytokines, as the signaling programs induced by adding pairs of parental cytokines simultaneously were dissimilar from those induced by the corresponding synthekines (*Figure 3—figure supplement 1*).

In order to determine whether the signaling programs activated by the synthekines differed from those activated by IL-2, IL-4 and IFN, we studied the activation of 120 different signaling molecules by phospho-flow cytometry in the CD4$^+$ T cell line Hut78 (*Figure 4*) (*Moraga et al., 2015*). Of the 120 molecules studied, twenty were activated by the ligands. The natural cytokine profiles were as expected: Super-2 strongly activated STAT5 and the PI3K pathways (*Figure 4A and B*); IL-4 stimulation robustly induced STAT6 and the PI3K pathway activation (*Figure 4A and B*); and IFN led to a strong activation of all STATs molecules (*Figure 4A and B*). When Hut78 cells were stimulated with the different synthekines, we could detect novel signaling programs engaged by these engineered ligands. Stimulation with SY1 LL (herein referred to as SY1) strongly activated STAT5 and STAT6, and also stimulated STAT1 and STAT3 to a lower extent (*Figure 4A and B*). In addition, this synthekine induced strong activation of the PI3K pathway (*i.e.* GSK3B, Akt, RPS6) (*Figure 4A*). SY2 stimulation resulted in an overall weaker signal than the other ligands, with preferential activation of STAT1 (*Figure 4A and B*). Moreover, the STAT activation ratios elicited by the cytokines and synthekines differed significantly, with SY1 exhibiting a STAT5/STAT6 preference and SY2 exhibiting a STAT1 preference (*Figure 4C*). Principal component analysis of the signaling programs elicited by genome-encoded cytokines and synthekines further confirm that synthekines activate distinct signaling programs and not only a subset of the original programs engaged by the parental cytokines (*Figure 4—figure supplement 1*).

## Cellular and signaling signatures induced by synthekines

We analyzed responses to synthekine treatment in 31 cell populations profiled from human PBMCs via mass cytometry (CyTOF) (*Bendall et al., 2011*) (*Figure 5A* and *Figure 5—figure supplement 1*). The native cytokines behaved as anticipated: Super-2 strongly activated STAT5, and also, to a lesser extent, activated STAT1, STAT3, Erk, and S6R, exhibiting a clear T cell preference (*Figure 5A*); IL-4 stimulation resulted in potent activation of STAT6 and a homogenous signaling footprint for T cells and monocytes, in agreement with the ubiquitous expression of the IL-4R$\alpha$ and $\gamma$c receptor subunits (*Figure 5A*). Stimulation with IFN promoted strong activation of STAT5 and STAT6 and weaker activation of STAT1 and STAT3, in agreement with previous observations (*van Boxel-Dezaire et al., 2010*) (*Figure 5A*). The IFN-induced STAT activation profile mapped into two different cell clusters, with T cells inducing stronger STAT5 and STAT6 activation and B cells and monocytes exhibiting strong STAT6 activation but weak STAT5 activation (*Figure 5A*). Cell signaling patterns elicited by synthekines diverged from those elicited by endogenous cytokines (*Figure 5A*). The SY1 synthekine induced strong STAT5 activation in T cells, but failed to activate signaling in NK, B cells, and monocytes (*Figure 5A*). The SY2 synthekine elicited weak signal activation of each signal effector studied in all cell populations, with a small bias towards STAT1 activation (*Figure 5A*).

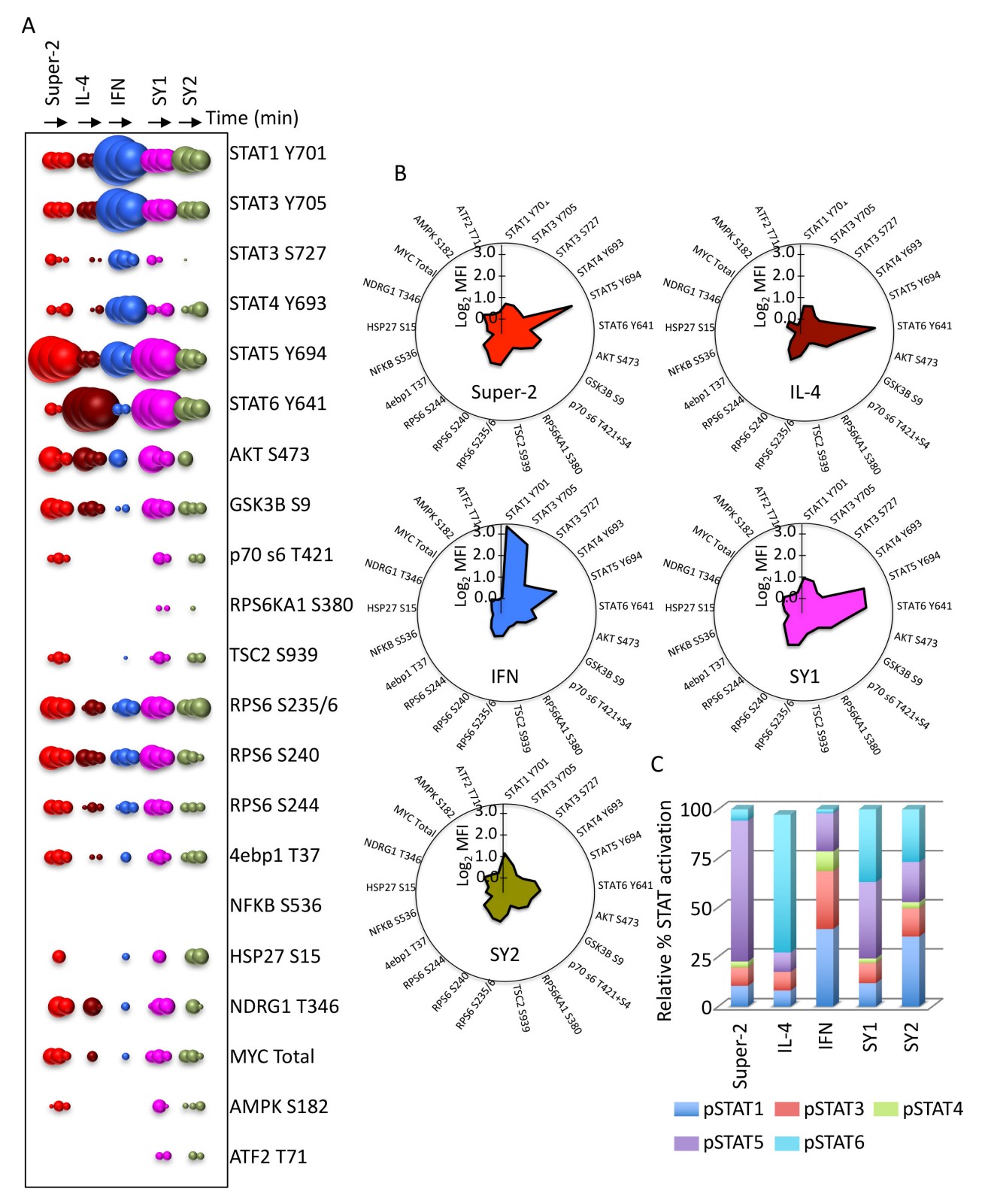

**Figure 4.** Synthekines activate different signaling programs than genome-encoded cytokines. (**A**) Bubble plot representation of the signaling pathways activated by the indicated ligands after stimulation for 15, 60 or 120 min in Hut78 T cells. The size of the bubble represents the intensity of the signal activated. (**B**) Filled radar representation of the signaling molecules activated by the genome-encoded cytokines and synthekines following 15 min stimulation in Hut78 cells. The signaling molecules activated by the ligands are shown on the perimeter of the circle and their respective activation

*Figure 4 continued on next page*

*Figure 4 continued*

potencies are denoted by the radius of the circle. The different shapes of the filled radar exhibited by the different ligands define their distinct signaling signatures. (C) Ratio of STAT activation by cytokines and synthekines after 15 min stimulation on Hut78 cells. Each column represents the total STAT activation by each ligand normalized to 100%. The relative activation potency of each STAT is corrected accordingly. The different distribution of STAT activation by the various ligands suggest differential STAT usages between genome-encoded cytokines and synthekines. Data (mean) are from two independent replicates.

The following figure supplement is available for figure 4:

**Figure supplement 1.** Unsupervised clustering of signaling programs engaged by cytokines and synthekines.

## Cytokine secretion profiles induced by synthekines

After ligand stimulation, secreted cytokine levels in the extracellular milieu are often used to define the nature of the immune response generated by a given cytokine (*Zhu et al., 2010*). We studied the cytokine secretion signatures induced by synthekines versus native cytokines. PBMCs were stimulated with IL-2, IL-4, IFN or the synthekines and the levels of 63 different analytes were measured after 24 hr of stimulation via bead-based immunoassay (*Figure 5B*). Super-2 and the two synthekines increased cytokine secretion, IFN had a neutral effect, and IL-4 reduced the amount of cytokine secreted by PBMCs (*Figure 5B*). More detailed analysis of the data revealed that, as expected, stimulation of PBMCs with Super-2 promoted secretion of high levels of LIF (*Umemiya-Okada et al., 1992*), IL-13 (*Hoshino et al., 1999*) and IFNγ (*Kasahara et al., 1983*) (*Figure 5B*). In addition, Super-2 resulted in secretion of IL-22, and CD40L by PBMCs (*Figure 5B*). Also consistent with previous reports, IFN stimulation induced secretion of IL-27 (*Greenwell-Wild et al., 2009*), while IL-4 stimulation led to down-regulation of cytokines secreted by resting PBMCs, with IFNγ being the most potently down-regulated cytokine (*Figure 5B*) (*Vercelli et al., 1990*). Stimulation profiles for the two synthekines differed from those induced by native cytokines. SY1 stimulation induced secretion of many cytokines: IL-17F, IL-27, IL-13, IL17A, IFNγ, BDNF, IL-23, FGFβ, PDGFBB, and ENA78, and SY2 stimulation led to marginal levels of Eotaxin, BDNF and PDGFBB secretion (*Figure 5B*).

## Synthekines dimerizing an RTK with a JAK/STAT receptor activate signaling

JAK/STAT cytokine receptors represent only a subset of single-pass transmembrane receptors that signal via dimerization-induced kinase activation. Receptor Tyrosine Kinases (RTKs) (e.g. EGFR [epidermal growth factor receptor], c-Kit, etc), represent another large family of dimeric cell-surface receptors that signal through trans-phosphorylation of their intracellular kinase domains (*Lemmon and Schlessinger, 2010*). We wondered if we could extend the scope of synthekines to include molecules that would compel heterodimerization and activation between a JAK/STAT cytokine receptor and an RTK.

To assess the possibility of JAK/STAT receptor cross-talk with an RTK, we fused the TM and ICD of EGFR to the ECD of IL-1R1 and transfected this construct together with our battery of IL-1R1AcP-cytokine receptors ICDs in Jurkat cells (*Figure 6A and B*). All ten cytokine-receptor/EGFR pairs expressed on the surface of Jurkat cells (*Figure 6—figure supplement 1A*). Stimulation with IL-1 resulted in variable degrees of phosphorylation of EGFR and, to a much lesser extent, STAT3 and STAT5 proteins (*Figure 6B*), suggesting that these cytokine and tyrosine kinase receptors are, in principle, capable of trans-phosphorylation when compelled through enforced proximity. This is consistent with prior studies showing that examples exist of such cross-talk can occur on natural cells (*Vignais et al., 1996*; *Grant et al., 2002*; *Jahn et al., 2007*; *Wang et al., 2013*). However, a caveat to these chimeric receptor studies is that overexpression of kinase-linked receptors can lead to aberrant, artifactual phosphorylation events. Therefore we sought to enforce heterodimerization of JAK/STAT and RTK-mediated receptors normally expressed on natural cells, in the absence of overexpression, using synthekine ligands.

We created a synthekine to compel dimerization of cKit, a tyrosine kinase receptor, and thrombopoietin receptor (TpoR), a JAK/STAT cytokine receptor. To design the synthekine bi-specific ligand, we identified sequences of antibodies that bound with high affinity to either cKit and TpoR ECD,

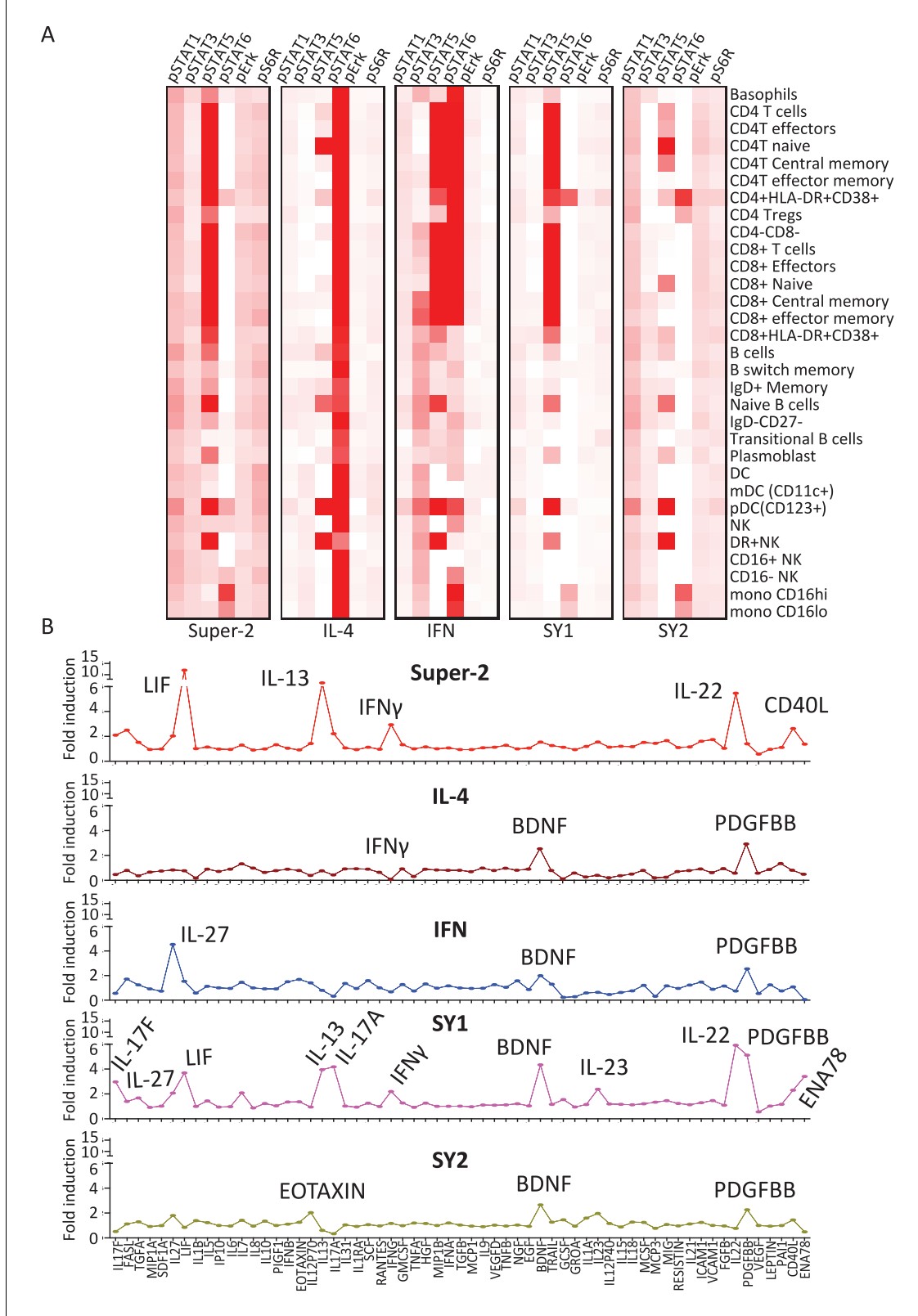

**Figure 5.** Synthekines elicit different cellular signatures and immune activities than genome-encoded cytokines. (**A**) Heat map representations of the activation levels of six signal effectors induced by saturating doses of the indicated ligands in 29 immune cell types profiled from PBMCs, as measured by mass cytometry (CyTOF). Data (mean) are from two independent replicates. (**B**) Detailed analysis of the secretion profiles of 63 cytokines from

*Figure 5 continued on next page*

*Figure 5 continued*

PBMCs stimulated with the indicated ligands. Cytokines that were secreted more than two fold above background are labeled. Data (mean ± SD) are from two independent replicates.

The following figure supplement is available for figure 5:

**Figure supplement 1.** Immune cell profiling to identify signaling signatures activated by cytokines versus synthekines.

respectively. We reformatted each of them as single-chain variable fragments (scFvs), and enforced their heterodimerization by fusing each with complementary acidic and basic leucine zippers (*Figure 6C*), expressing and purifying them from insect cells, and applying them to the acute mega-karyoblastic leukemia Mo7e cells, which are known to express cKit and TpoR. The SY4 and SY5 syn-thekines both induced modest phosphorylation of cKit in Mo7e cells over background, but only SY5 induced detectable phosphorylation of TpoR-associated JAK2 over background, albeit very weakly (*Figure 6D*). SY5 induced measurable Erk activation (50% Emax compared to the native ligand, stem cell factor [SCF]) but only weak STAT5 activation, which is consistent with the apparent asymmetric activation of cKit over TpoR (*Figure 6E*). Indeed, inhibition of JAK2 using a JAK2 small molecule inhibitor resulted in loss of Erk activation by SY5, suggesting that the signaling program engaged by these synthekines relies, at least in part, on JAK2 activity (*Figure 6—figure supplement 1B*). Thus, there appears to be asymmetry in the efficiency of trans-phosphorylation within the TpoR/cKit heterodimer.

We performed a high throughput phospho-flow cytometry-based study (*Moraga et al., 2015*) to analyze the signaling response of 120 signaling molecules in stimulated Mo7e cells. As shown in *Figures 7A* and 54 of the 120 different signaling molecules were activated above a significance threshold by the different ligands. Interestingly the signaling signature elicited by SY5 appeared to evoke qualitatively different outputs than SCF, TPO or the combination of the two ligands; depend-ing on the pathway effector studied (*Figure 7B and C*). Collectively, the chimeric receptor and syn-thekine studies show that, although inefficient compared to their natural ligands, JAK/TYK and RTK-mediated signaling receptors are capable of cross-talk when compelled into heterodimeric com-plexes by synthekines, which is consistent with prior studies suggesting that JAK and the RTK com-ponents are capable of phosphorylating one another's natural substrates (*Vignais et al., 1996*; *Grant et al., 2002*; *Jahn et al., 2007*; *Wang et al., 2013*). More extensive exploration of this obser-vation with optimized ligands tuned to maximize signaling output will be required to further clarify this observation.

## Discussion

In this study, we sought to expand the scope of kinase-linked dimeric receptor signaling on natural cells using synthetic ligands, which we term Synthekines, to compel formation of non-natural recep-tor dimers. Synthekines can be loosely analogized to 'orphan' cytokines since their physiological functions are not known. However an important distinction from true orphan ligands is that synthe-kines have known, pre-defined receptors, which greatly limits the potential functional scope of their actions to the given cell types that express their receptors. This approach can exploit the full combi-natorial potential of JAK/TYK/STAT, and RTK signaling through the creation of artificial receptor dimers on natural cells. A compelling rationale for our exploring this approach is that, despite their immunotherapeutic potential, relatively few cytokines are useful clinically, due in large part to their pleiotropy and off-target effects (*Andrews et al., 2006*; *Borden et al., 2007*; *Donohue and Rosen-berg, 1983*; *Spangler et al., 2015*). In recent years, cytokine variants have been engineered with more defined activities and reduced toxicity (*Juntilla et al., 2012*; *Levin et al., 2012*, *2014*). How-ever, an intrinsic limitation to this approach is that engineered cytokines exhibit a subset of activities within the bioactivity space occupied by the parental cytokine. Our proof-of-concept experiments show that activation of distinct signaling programs and by extension, immune activities, can be accomplished through engineering of synthetic cytokines (synthekines) that dimerize non-natural cytokine receptor pairs. This approach offers the opportunity for generating 'designer' ligands to specifically target biological processes relevant to health and disease.

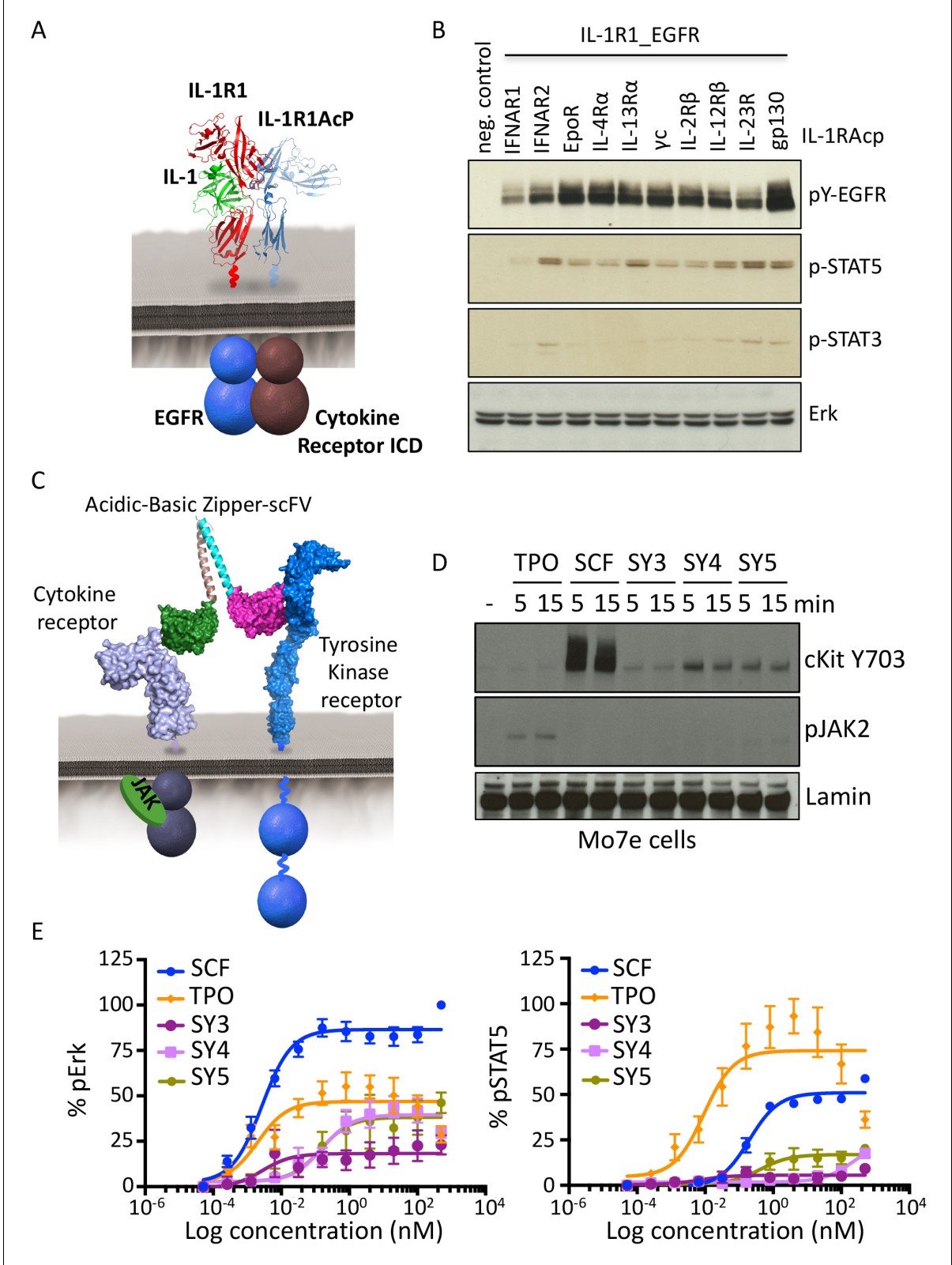

**Figure 6.** Synthekines dimerizing a cytokine receptor and a tyrosine kinase receptor activate signaling. (**A**) Schematic representation of the IL-1-mediated complexation of IL1-R1-EGFR and IL-1R1AcP-cytokine receptor chimeras. (**B**) Phospho-EGFR (pY EGFR), pSTAT3 and pSTAT5 levels measured by western blot analysis in IL-1-activated Jurkat cells expressing the indicated chimeric receptor pairs. Total levels of Erk are presented as a loading control. The western blot presented is a representative example of two independent experiments. (**C**) Layout and complex formation by a synthekine

*Figure 6 continued on next page*

*Figure 6 continued*

dimerizing a cytokine receptor and a tyrosine kinase receptor. Two scFvs binding a cytokine receptor and a tyrosine kinase receptor respectively are genetically fused to acidic or basic leucine zippers, resulting in a new molecule able to form a heterodimeric receptor complex that does not exist in nature. (D) Phospho cKit Y703 and pJAK2 levels measured by western blot in Mo7E cells after stimulation with synthekines that dimerize TpoR and cKit (SY3, SY4 and SY5) for the indicated time periods. Total levels of Lamin are presented as a loading control. The western blot presented is a representative example of two independent experiments. (E) Erk (left panel) and STAT5 (right panel) phosphorylation activated by 10 min stimulation with the indicated doses of SCF, TPO, or the indicated synthekines in Mo7e cells, as measured by flow cytometry. Data (mean ± SD) are from three independent replicates.

The following figure supplement is available for figure 6:

**Figure supplement 1.** Functional characterization of tyrosine kinase receptor/cytokine receptor dimerizing synthekines.

Our data show that synthekines activate distinct and novel signaling programs and induce secretion of new cytokine signatures by stimulated PBMCs. However, we still lack a clear mechanistic understanding for how specificity of synthekine signaling and activity is achieved. There are several factors that could account for this specificity. First, while most synthekines elicit some signals that partially resemble those of the parent cytokines, the ratio of activated STATs differs significantly between the parent cytokines and synthekines. For instance, SY1 elicits a STAT6>STAT5>STAT1>STAT3 pattern instead of the STAT6>STAT1>STAT3>STAT5 pattern seen with IL-4 plus IFN, while SY2 elicits a STAT1>STAT6>STAT5>STAT3 pattern instead of the STAT6>STAT5>STAT3>STAT1 pattern seen with IL-4 plus IL-2. The physiological relevance of different activated STAT ratios remains to be fully understood; however several reports have shown that changes in STAT activation ratios can alter cytokine-induced biological responses (*Gil et al., 2012*; *Sharif et al., 2004*; *Thomas et al., 2011*). Second, given that synthekines induce dimerization of non-naturally occurring cytokine receptor pairs, it is conceivable that they can change the abundance of STAT heterodimers and possibly even induce formation of novel STAT heterodimer pairings, which could result in the induction of completely novel gene expression programs and activities.

A limitation of our study is that we focused on human receptors and therefore have not carried out in vivo experiments, but the results we present provide a compelling rationale for exploration of synthekines in mouse systems and disease models. For investigation in mouse systems, suitable high affinity binding modules to the receptor ECDs will need to be generated in order to create bi-specific entities. The physiological effects of synthekines will be no less complex than natural cytokines, requiring in-depth study, but knowing the activities of the parent receptor chains used to form the non-natural dimer could lend clues to potential new activities by the synthekine. For example, we expect that in some cases the physiological effects and disease applications could be similar or related to those of one of the parent receptor chains, while in other cases entirely distinct.

The synthekine design paradigm encompasses several critical considerations: (1) Selection of two cytokine receptor subunits simultaneously expressed in the same cell. Cellular response to cytokines is tightly regulated by surface expression patterns of cytokine receptor subunits (*Levin et al., 2012*; *Moraga et al., 2009*; *Wang et al., 2009*). Thus, there are many cytokine receptor pair combinations that, although compatible with signaling in principle, would not have physiological relevance due to the lack of a naturally occurring cell subset that simultaneously expresses the two receptors subunits. (2) Selection of the cytokine receptor subunit types to be dimerized by synthekines. From our chimeric receptor study, we infer that most cytokine receptor pair combinations will activate signaling to some extent. However, other parameters such as structural properties may influence the degree and nature of signaling activation. For example, cytokine receptors can be subdivided into two classes based on ICD length. Receptors with long ICDs often bind their ligands with high affinity, pair with JAK1 or JAK2, encode for STAT binding sites, and drive signal activation (*Murray, 2007*; *Wang et al., 2009*). In contrast, receptors with short ICDs often bind their ligands with lower affinity, pair with TYK2 or JAK3, and minimally contribute to STAT recruitment and activation (*Murray, 2007*; *Wang et al., 2009*). Interestingly, many receptor pairs that did not activate signaling in our chimeric receptor study comprised short ICD receptors, suggesting that synthekines dimerizing two short ICD receptor subunits would elicit weaker and less diverse signal activation programs than those dimerizing long ICDs receptor subunits. In addition, our results show that synthekines dimerizing

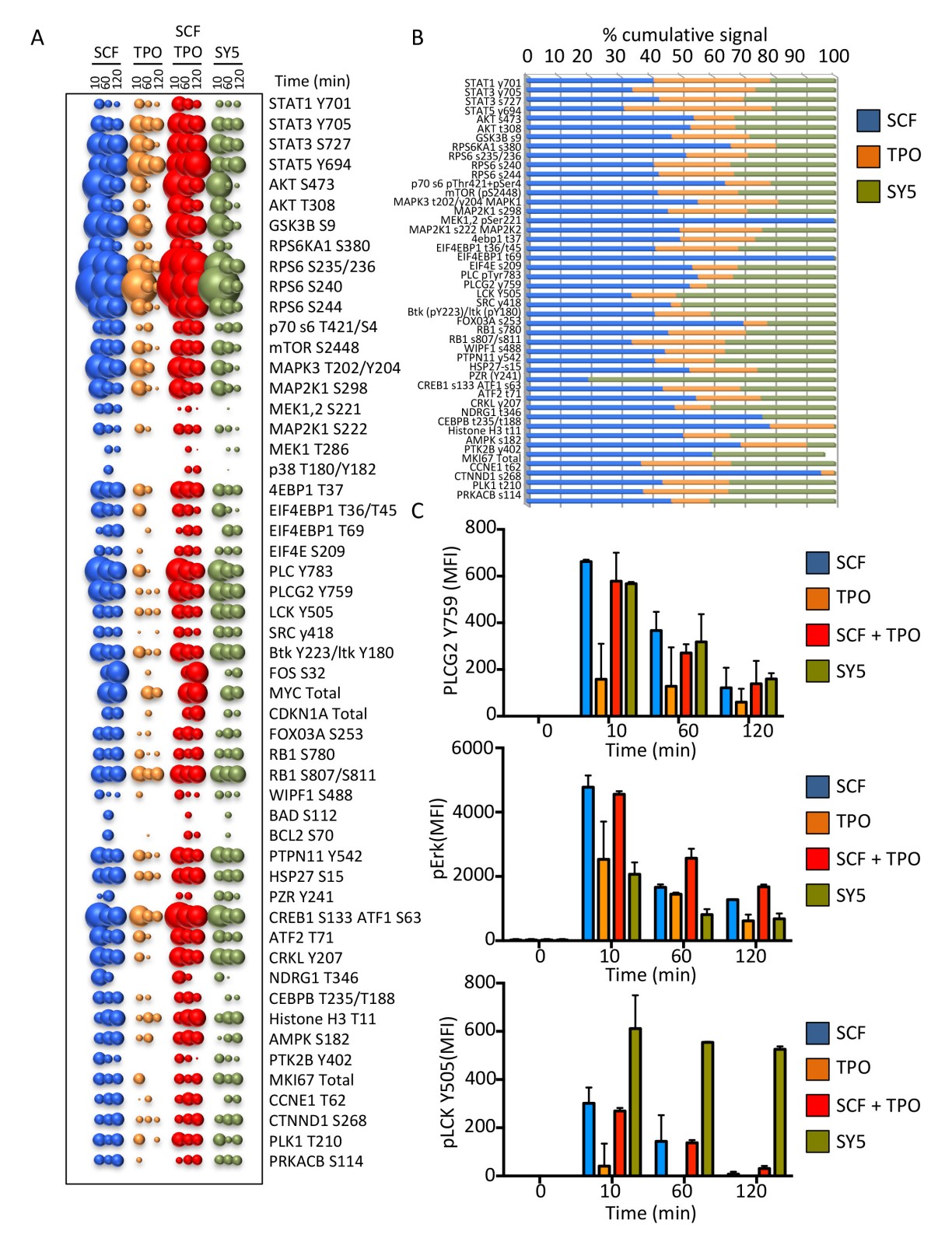

**Figure 7.** Synthekines dimerizing a cytokine receptor and a tyrosine kinase receptor activate different signaling programs than their natural ligands. (**A**) Bubble plot representation of the signaling pathways activated by the indicated ligands after stimulation for 10, 60 and 120 min in Mo7e cells. The size of the bubble represents the intensity of the signal activated. (**B**) Stack column representation of the signaling molecules engaged by SCF, TPO and SY5 after 10 min stimulation in Mo7e cells. For each molecule, the combined activation of the three ligands was normalized to 100% and the

*Figure 7 continued on next page*

*Figure 7 continued*

relative contribution of each ligand was corrected accordingly. Some molecules were better activated by SCF, others were better activated by TPO, and yet others were better activated by SY5. Data presented in panel **A** and **B** represents the mean value of two independent experiments performed in triplicate. (**C**) pPLCG2, pErk, and pLCK levels induced by the indicated ligands in Mo7e cells after 10, 60 and 120 min stimulation. Data (mean ± SD) are from three independent replicates.

receptors from two different cell surface receptor families (specifically the cytokine receptor and the tyrosine kinase receptor families) can activate signaling, albeit low efficiency, in accord with previous findings that hinted at cross-talk between these receptor families (*Wang et al., 2013*).

An important aspect of the synthekine technology is that they dimerize cytokine receptor pairs in a defined 1:1 molecular entity, which enables clear attribution of the signaling pathway to a specific receptor dimer. Formation of a molecularly-defined surface complex by an engineered ligand is vital for characterizing the signaling and phenotypic programs activated by these ligands. Previous studies linked together two fully functional cytokines and generated ligands that could form multiple independently functioning or mixed receptor complexes (*Ng and Galipeau, 2015*). In contrast, the targeted synthekine approach using linked antagonist cytokine mutants or antibody-based binding modules that can only bind to one receptor subunit, allows one to interrogate the activity of specific receptor dimer pairs of defined composition and stoichiometry.

A consistent finding from our study was that the engineered synthekines were relatively inefficient at activating signaling compared to the parent genome-encoded cytokines; at best we could detect 60% of the signaling amplitude induced by IL-2, IL-4, or IFN. The synthekines that evoked signaling from the cKit/TpoR heterodimer exhibited even weaker activation properties. One possible explanation for this observation is that the architecture of the cytokine-receptor complex appears to be a determinant of signal potency (*Moraga et al., 2015*). It is thus possible that the receptor binding topology induced by the engineered synthekines in both cases is suboptimal and that signaling strength could be improved by altering the construction of these molecules in order to optimize the receptor dimer topology. Future studies will focus on the identification of suitable molecular scaffolds that improve the efficiency of synthekine activation.

## Materials and methods

### Protein expression and purification

Human IL-4, Super-2, IFN, dominant negative cytokines, and synthekines were expressed and purified using a baculovirus expression system, as described in (*Laporte et al., 2005*). The sequence for the Super-2 variant of IL-2 is provided in (*Levin et al., 2012*). The SY1 SL and LL synthekines were generated by genetically fusing the IL-4DN and IFNDN proteins via a single (SY1 SL) or double (SY1 LL) Gly$_4$Ser linker. The SY2 synthekine was generated by genetically fusing the IL-2DN and IL-4DN proteins via a Gly$_4$Ser linker. IL-4DN was generated by introducing the previously described R121D/Y124D mutations on site II, which disrupt binding to common gamma chain (*Wenzel et al., 2007*). IFNDN was generating by disrupting the binding to IFNAR1 by introducing the mutations F63A and R120E on the IFNω-IFNAR1 binding interface. IL-2DN (also known as IL-2 RETR) was previously described in *Mitra et al. (2015)*. Single-chain variable fragments (scFvs) used for engineering SY3, SY4 and SY5 were analogously expressed and purified in the baculovirus system via transfer of their variable regions into the pAcGP67A vector (BD Biosciences, San Jose, CA) with an N-terminal gp67 signal peptide and a C-terminal hexahistidine tag. scFvs were expressed with the variable heavy (V$_H$) and variable light (V$_L$) chains separated by a twelve-amino acid (Gly$_4$Ser)$_3$ linker fused either to acidic or basic leucine zippers for dimerization. All proteins contained C-terminal hexahistidine tags and were isolated by nickel chromatography and further purified to >98% homogeneity by size exclusion chromatography on a Superdex 200 column (GE Healthcare, UK), equilibrated in 10 mM HEPES (pH 7.3) and 150 mM NaCl.

## Chimeric receptor generation

In order to generate the 10 × 10 signaling matrix, the ICDs of the 10 different parental cytokine receptors were fused with the IL-1R1 and IL-1R1Acp ECDs. In the IL-1R1 ECD format, the nucleotide sequence encoding the HA-tag was inserted between the end of the native signal sequence and the first residue of the IL-1R1 ECD. Each ICD was fused to the 3′ end of IL-1R1 sequence. The IL-1R1Acp chimeras were cloned in the same manner except the V5-tag was used. The boundaries of the mature proteins and transmembrane spans were delineated using the SignalP (http://www.cbs.dtu. dk/services/SignalP/) and TMHMM (http://www.cbs.dtu.dk/services/TMHMM/) webservers. The DNA sequence used for IL-1R1 was codon optimized for expression in Homo sapiens as the organism (jcat.de) and synthesized (Integrated DNA Technologies). The chimeric receptors were cloned into the pcDNA3.1 + vector (Invitrogen) using the NheI and KpnI restriction sites (NEB).

| Receptor (ECD) | Signal sequence | Tag | ECD start | ECD end | NCBI reference seq |
|---|---|---|---|---|---|
| IL-1R1 | IL-1R1 | HA | 1 | 315 | NM_000877.2 |
| IL-1RAcp | IL-1RAcp | V5 | 1 | 335 | NM_002182.3 |
| Receptor (ICD) | Cloning Start | TM Start | NCBI Reference Seq | | |
| EPO-R | 223 | 226 | NM_000121.3 | | |
| gp130 | 595 | 598 | BC117402.1 | | |
| IFNAR1 | 408 | 411 | NM_000629.2 | | |
| IFNAR2 | 214 | 217 | U29584.1 | | |
| IL-12Rb | 218 | 221 | NM_005535.1 | | |
| IL-13Ra1 | 219 | 222 | NM_001560.2 | | |
| IL-23R | 218 | 221 | NM_144701.2 | | |
| IL-2Rb | 215 | 218 | NM_000878 | | |
| IL-4Ra | 218 | 221 | NM_000418.3 | | |
| γc | 219 | 222 | NM_000206.2 | | |

## Tissue culture

Jurkat (CVCL_0367) and HUT78 (CVCL_0337) cell lines were acquired from the ATCC cell repository. STR profiles were performed by ATCC to ensure identity of the lines. Mo7e (CVCL_2106) cells were purchased from the DSMZ repository. Multiplex PCR of minisatellite markers performed by DSMZ revealed a unique DNA profile. All cell lines were negative for mycoplasma. Jurkat cells were cultured in DMEM complete medium (DMEM medium supplemented with 10% FBS, 2 mM L-glutamine, and penicillin-streptomycin (Gibco, Waltham, MA)). Hut78 cells were cultured in RPMI complete medium (RPMI 1640 medium supplemented with 10% FBS, 2 mM L-glutamine, and penicillin-streptomycin (Gibco)). Mo7e cells were cultured in IMEM complete media (IMEM medium supplemented with 10% FBS, 2 mM L-glutamine, 10 nM GM-SCF and penicillin-streptomycin (Gibco)). Prior to stimulation, Mo7e cells were starved overnight in modified growth medium lacking FBS and GM-CSF. All cell lines were maintained at 37°C in a humidified atmosphere with 5% $CO_2$.

## Hut78 and Mo7e intracellular signaling studies

Approximately $3 \times 10^5$ Hut78 or Mo7e cells per well were placed in a 96-well plate, washed with PBSA buffer (phosphate-buffered saline (PBS) pH 7.2, 1% BSA), and re-suspended in PBSA containing serial dilutions of the indicated ligands. Cells were stimulated for the prescribed time at 37°C and immediately fixed by addition of formaldehyde to 1.5% followed by incubation for 10 min at room temperature. Cells were then permeabilized with 100% ice-cold methanol for 30 min at 4°C. The fixed and permeabilized cells were washed twice with PBSA and incubated with fluorescently-labeled detection antibodies diluted in PBSA for 1 hr at room temperature. Fluorophore-conjugated pSTAT3 (AB_647098), pSTAT5 (AB_11154412) and pSTAT6 (AB_399883) antibodies were purchased from BD Biosciences (San Jose, CA). pSTAT1 (AB_10860764), pErk (AB_10695308),

pcKit (AB_1147635), pEGFR (AB_331701) antibodies were purchased from Cell Signaling Technology. Cells were then washed twice in PBSA buffer and mean fluorescence intensity (MFI) was quantified on an Accuri C6 flow cytometer. Dose-response curves were fitted to a logistic model and $EC_{50}$ values were computed in the GraphPad Prism data analysis software after subtraction of the MFI of unstimulated cells and normalization to the maximum signal intensity induced by wild-type cytokine stimulation.

## Peripheral blood mononuclear cell (PBMC) isolation from human whole blood

Peripheral blood mononuclear cells (PBMCs) were isolated from human whole blood (Stanford Blood Bank) using a gradient of Ficoll-Paque Plus (GE Healthcare) according to the manufacturer's protocol. Freshly isolated PBMCs were used for both mass cytometry studies and bead-based immunoassays. Prior to stimulation, PBMCs were rested at 37°C, 5% $CO_2$ for 1 hr in RPMI complete medium.

## Mass cytometry immune cell signaling analysis

This assay was performed in the Human Immune Monitoring Center at Stanford University. Freshly isolated PBMC were seeded in 96-well plates at $5 \times 10^5$ cells per well and stimulated with serial dilutions of the indicated ligands in RPMI complete for 20 min at 37°C, 5% $CO_2$. Cells were then fixed via 10 min incubation in paraformaldehyde (1.5% final concentration) at room temperature. Cells were washed and resuspended in CyFACS buffer (PBS supplemented with 2% BSA, 2 mM EDTA, and 0.1% sodium azide) containing the metal-chelating polymer-labeled anti-surface antigen antibodies for 30 min at room temperature. Antibodies were labeled from purified unconjugated, carrier-protein-free stocks from BD Biosciences, Biolegend (San Diego, CA), or Cell Signaling Technology (Danvers, MA) and the polymer and metal isotopes were from DVS Sciences. Cells were washed once in CyFACS buffer and then permeabilized overnight in methanol at −80°C. The following day, cells were washed once in CyFACS buffer and resuspended in CyFACS buffer containing the metal-chelating polymer-labeled anti-intracellular antigen antibodies for 30 min at room temperature. Cells were washed twice in PBS (phosphate-buffered saline pH 7.2), resuspended in iridium-containing DNA intercalator (1:200 dilution in PBS, DVS Sciences) and incubated on ice for 20 min. The cells were then washed three times in MilliQ water and then diluted in a total volume of 700 uL in MilliQ water before injection into the CyTOF instrument (DVS Sciences, Sunnyvale, CA).

Data analysis was performed using FlowJo (CyTOF settings) by gating on cells based on the iridium isotopes from the intercalator, then on intact singlets based on plots of one intercalator iridium isotope vs. cell length, followed by cell subset-specific gating. Signal intensity for each condition is reported as 90th percentile intensity minus that of an unstimulated control sample. Heat maps of the response to the maximum concentration of each treatment were generated using the TM4 microarray software suite (Dana-Farber Cancer Institute) (*Saeed et al., 2003*), with signal intensities normalized to the maximum signal effector response in each cell type. Two independent replicates of mass cytometry experiments were performed with similar results obtained.

## Bead-based immunoassay cytokine secretion studies

This assay was performed in the Human Immune Monitoring Center at Stanford University. Freshly isolated PBMCs ($1.5 \times 10^5$ per well) were stimulated with the indicated ligands in the presence of 1 µg/mL phytohaemagglutinin (PHA) in RPMI complete and incubated for 24 hr at 37°C, 5% $CO_2$. Cells were then pelleted via centrifugation and supernatants were harvested for bead-based immunoassay analysis using the Luminex platform (Luminex Corporation, Austin, TX). Human 63-plex kits were purchased from eBiosciences/Affymetrix and used according to the manufacturer's recommendations with modifications as described below. Briefly: Beads were added to a 96 well plate and washed in a Biotek ELx405 washer. Harvested supernatants were added to the plate containing the mixed antibody-linked beads and incubated at room temperature for 1 hr followed by overnight incubation at 4°C with shaking. Cold and room temperature incubation steps were performed on an orbital shaker at 500–600 rpm. Following the overnight incubation, plates were washed in a Biotek ELx405 washer and biotinylated detection antibody was added for 75 min at room temperature with shaking. Plates were washed again as above and streptavidin-PE (Invitrogen, Waltham, MA) was added. After incubation for 30 min at room temperature, a final wash was performed as above and reading buffer was

added to the wells. Plates were analyzed on a Luminex 200 instrument with a lower bound of 50 beads per sample per cytokine. Custom assay Control beads (Radix Biosolutions, Georgetown, TX) were added to all wells. Each sample was measured in duplicate and raw MFI was averaged from the two replicates. Results are presented as fold change in MFI of treated cells relative to control cells stimulated with PHA only.

### Primity bio pathway phenotyping

Hut78 or Mo7e cells were stimulated with saturating concentrations of the indicated ligands for 15, 60 and 120 min and fixed with 1% PFA for 10 min at room temp. The fixed cells were prepared for antibody staining according to standard protocols (*Krutzik and Nolan, 2003*). Briefly, the fixed cells were permeabilized in 90% methanol for 15 min. The cells were then stained with a panel of antibodies specific to the markers indicated (Primity Bio Pathway Phenotyping service) and analyzed on an LSRII flow cytometer (BD Biosciences). The $\log_2$ ratio of the MFI of the stimulated samples divided by the unstimulated control samples were calculated as a measure of response.

### Western blot analysis

Cells were lysed in 1% NP-40 lysis buffer and 30 µg protein was analyzed as described in (*Marijanovic et al., 2007*). The following polyclonal antibodies were used: anti-phospho Tyk2, anti-phospho STAT1; anti-phospho STAT2; anti-phospho STAT3; anti-phospho STAT4; anti-phospho STAT5; anti-phospho STAT6; anti-phospho EGFR; anti-phospho cKit pY703 (Cell Signaling Technology). Signal was revealed with the ECL enhanced chemiluminescence Western blotting reagent Western Lightning Chemiluminescence Reagent Plus (PerkinElmer, Waltham, MA).

### Electroporation

$15–20 \times 10^6$ Jurkat cells maintained at densities between $0.5–1.0 \times 10_6$ cells/ml were washed twice with RPMI medium (sterile) and resuspended in 0.25 ml of Ingenio electroporation solution (Mirusbio, Madison, WI). 5–30 ug DNA (not exceeding 15% of the total volume) was added to the resuspended Jurkat cells and the mixture was transferred to a 4 mm gap cuvette (Invitrogen) and incubated for 15–20 min room temperature. Cells were electroporated in a Biorad (Hercules, CA) electroporator set at 0.28 kV and 960 µF. After electroporation, cells were transferred to prewarmed media (without Pen/Strep) and cultured normally. Protein expression was monitored after 24 hr.

### Cell surface receptor staining

Surface receptor levels were monitored as described in (*Marijanovic et al., 2007*), using fluorescently-labeled monoclonal antibodies specific for the HA and V5 tags (Cell Signaling Technology). Electroporated Jurkat cells were resuspended in cold PBS containing 3% fetal calf serum and incubated with the indicated antibodies for 1 hr. Samples were analyzed on an Accuri C6 flow cytometer (BD Biosciences).

## Acknowledgements

We thank R Fernandez, Y Rosenberg-Hasson, and H Maecker of the Stanford Human Immune Monitoring Facility for assistance with mass cytometry studies and bead-based immunoassays. JLM is supported by NIH award K01CA175127. JBS is the recipient of a Leukemia & Lymphoma Society Career Development Program fellowship. This work was funded by NIH-RO1-AI51321, Ludwig and Mathers Foundations (to KCG), and KCG is an Investigator of the Howard Hughes Medical Institute.

## Additional information

### Funding

| Funder | Grant reference number | Author |
|---|---|---|
| Leukemia and Lymphoma Society Career Development Program fellowship | | Jamie B Spangler |

| National Institutes of Health | K01CA175127 | Juan L Mendoza |
| National Institutes of Health | RO1-AI51321 | Milica Gakovic |
| Howard Hughes Medical Institute | | K Christopher Garcia |

The funders had no role in study design, data collection and interpretation, or the decision to submit the work for publication.

## Author contributions

IM, Conceptualization, Data curation, Formal analysis, Investigation, Methodology, Writing—original draft, Project administration, Writing—review and editing; JBS, JLM, Conceptualization, Data curation, Formal analysis, Investigation, Methodology, Writing—original draft, Writing—review and editing; MG, Formal analysis, Investigation, Methodology, Writing—review and editing; TSW, PK, Methodology; KCG, Conceptualization, Data curation, Formal analysis, Supervision, Funding acquisition, Investigation, Writing—original draft, Project administration, Writing—review and editing

## Author ORCIDs

Ignacio Moraga, http://orcid.org/0000-0001-9909-0701
Jamie B Spangler, http://orcid.org/0000-0002-1664-2038
Tom S Wehrman, http://orcid.org/0000-0003-2275-4551
K Christopher Garcia, http://orcid.org/0000-0001-9273-0278

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
