## [Decision Letter]

Thank you for submitting your article "Synthekines: surrogate cytokine agonists that elicit new signals through non-natural receptor dimers" for consideration by *eLife*. Your article has been reviewed by two peer reviewers, and the evaluation has been overseen by a Reviewing Editor and Tadatsugu Taniguchi as the Senior Editor. The following individual involved in review of your submission has agreed to reveal his identity: Peter Murray.

The reviewers have discussed the reviews with one another and the Reviewing Editor has drafted this decision to help you prepare a revised submission. As you can see, the reviewers are essentially quite supportive to accept this paper.

Summary:

In this novel and innovative study, Moraga et al. describe the engineering and characterization of recombinant cytokines and cytokine receptors capable of engaging pre-defined (and unnatural) receptor pairings. The authors present new design strategies and an expansive multi-pronged approach that has both conceptual and practical merits. Their work makes important points about fundamental mechanisms of cytokine signalling and will open new areas of cytokine receptor signalling such as the potential deconvolution of the diversity in Stat3-mediated signalling between many receptors present on the same cell but eliciting very different responses. The creative approach of the authors also ushers in a new class of molecule, the 'synthekine', that will certainly be useful as a research tool and perhaps as a clinical agent. The work is extensive, elegant and carefully and comprehensively done. Some technical and conceptual issues that remain to be addressed are identified below.

Essential revisions:

1) Despite the extensive quantitative and qualitative evidence for the signalling cascades downstream of synthekines, it remains unclear how and why synthekines generated by the fusion of two cytokines diverge from their parents. The potential unique downstream cellular consequences are also not thoroughly assessed. Additional data addressing these issues would be helpful, possibly along the directions indicated in 1c) and 1d) below. Alternatively, the manuscript could be more explicitly presented as a valuable exercise in synthetic biology. While additional data would be useful, they are not essential, but the issues raised below, particularly in sections 1a) and 1b), should be explicitly discussed.

a) A central question is: do synthekines have unique biological properties or just manifest the combined, yet attenuated, activities of their parent cytokines? The distinction is crucial for understanding mechanism of action. On one hand, synthekines appear to behave like weak agonists; the STAT profile of cells exposed to SY1 mostly resembles that of cells exposed to low doses of IL-4 and IFN, while the STAT profile of cells exposed to SY2 mostly resembles that of cells exposed to low doses of IL-4 and IL-2 (Figure 3—figure supplement 1). Mass cytometry data for other downstream signalling pathways tends to support this view – most analytes are either unchanged or reduced compared to parent cytokines (Figure 4) – and the authors acknowledge such quantitative effects. On the other hand, they also provide evidence for qualitative differences. In particular, the ratio of STATs activated by synthekines appears to be distinct from that of parent cytokines; SY1 elicits a STAT6>STAT5>STAT1>STAT3 pattern instead of the STAT6>STAT1>STAT3>STAT5 pattern seen with IL-4 plus IFN, while SY2 elicits a STAT1>STAT6>STAT5>STAT3 patterns instead of the STAT6>STAT5>STAT3>STAT1 pattern seen with IL-4 plus IL-2. However, better understanding of downstream consequences is necessary to confirm that these changes translate to unique functions. The authors offer one set of experiments where they compare effects of synthekines and parent cytokines on PBMC cytokine production but the observed differences are subtle and the data are difficult to interpret vis-a-vis what we know about the parent cytokines. For instance, why does SY1 promote IL-17 production when its dominant signaling moieties, STAT6, STAT5 and STAT1, are all thought to inhibit this process (Yang et al., Nature Immunology 2011; Harrington et al., Nature Immunology 2005)? Why does it promote IL-13 production when its ability to activate STAT6, a major driver of this cytokine, is greatly diminished? And, in a related question, why does IL-4 fail to induce IL-13 despite unbridled STAT6 activation? It also bears noting that this experiment strongly supports the idea for SY2 as just a weak agonist; IL-2-driven effects are completely lost, IL-4-driven changes are diminished and there is only one unanticipated analyte, eotaxin, which is known to be regulated by IL-4 in a STAT6-dependent manner (Hoeck et al., JI 2001).

b) If we consider that the engineered receptors and synthekines presented in this paper drive juxtaposition of cytokine receptors that are not normally associated with one another, then the issue of STAT heterodimers becomes highly relevant. Certain cytokine receptors are already known to elicit heterodimer formation (Delgoffe & Vignali, JAKSTAT 2013) and it is possible that the unnatural receptor combinations described here could increase/decrease their prevalence or even form previously unseen species, such as STAT1/STAT6 heterodimers downstream of SY1, thus providing a molecular basis for unique functions.

c) Flow cytometry based assays may be more suitable for assessing synthekine function. The authors could compare expression of proteins that are known to be regulated by a particular STAT (e.g. LIF for STAT5, GATA-3 for STAT6, CD274 for STAT1, IL-21 for STAT3) in strictly defined PBMC populations (i.e. CD4^+^ T cells, monocytes) that have been treated with synthekines or escalating doses of one or both parent cytokines (as in Figure 3—figure supplement 1). In addition, since each of the parent cytokines is known to impact helper T cell differentiation (i.e. Th1 vs. Th2 vs. Th17), this may also be a useful model for assessing cellular effects of synthekines. Alternatively, defined PBMC populations could be sorted, treated with synthekines and/or parent cytokines and expression of known STAT-regulated genes (e.g. interferon signature genes) measured by PCR or, ideally, by microarray or RNA-seq.

d) Another possibility is whether synthekines can antagonize their parent cytokines in terms of signaling and/or downstream effects. Thus, experiments where synthekines are added to cells together with one or both parents would be informative.

2) There is also a lack of cohesiveness between the three principal components of the manuscript – the first two components (chimeric receptors and synthekines) clearly inform one another but the third section (about cytokine-RTK fusions) seems tangential. The latter constructs are certainly interesting from a basic science perspective – in fact, the data presented illustrate that Jaks can phosphorylate RTKs (at least EGFR and KIT) but not vice versa – and exciting in terms of therapeutic potential, but some attempt at improved integration of the manuscript would be desirable.

---

## [Author Response]

*Essential revisions:*

*1) Despite the extensive quantitative and qualitative evidence for the signalling cascades downstream of synthekines, it remains unclear how and why synthekines generated by the fusion of two cytokines diverge from their parents. The potential unique downstream cellular consequences are also not thoroughly assessed. Additional data addressing these issues would be helpful, possibly along the directions indicated in 1c) and 1d) below. Alternatively, the manuscript could be more explicitly presented as a valuable exercise in synthetic biology. While additional data would be useful, they are not essential, but the issues raised below, particularly in sections 1a) and 1b), should be explicitly discussed.*

We appreciate the careful analysis of our manuscript, and are in agreement with the reviewer’s and editor’s comments, with minor exceptions. This new approach raises many unanswered questions regarding mechanism and function that we intend to pursue in future studies. Our goal in this initial study was to 1- introduce the synthekine ligand engineering concept and feasibility, and 2- to demonstrate the surprising extent, and potential, of ligand-driven cross-talk between diverse kinase-linked receptors on natural cells. This latter point is important in that, at some point in the future, appropriate synthekine ligands could potentially be used in animals/humans as therapeutics. We understand the reviewers’ analogy of our work to a synthetic biology study, and we have revised our manuscript accordingly. However we think that the near-term biological implications of synthekines exceed those of many synthetic biology studies on contrived model systems that have little apparent biological relevance. Thus, in this spirit, we have modified the manuscript throughout (including title, Abstract and Introduction) to more closely hue to the ‘high level’ conceptual and technological advances, which is that the potential cross-talk of kinase-linked receptors can be exploited using synthetic bi-specific ligands. We have been careful not to stray far from this theme into functional speculations at this early stage.

We would like to point out several important limitations to our study that may help give the reviewers a context for our responses:

First, we made synthekines that act on human cells, and are not cross-reactive with mouse receptors, which limits us to in vitro experiments. We attempted to use a comprehensive battery of assays as to analyze the activity of the synthekines for this reason and make the point that these new entities exhibit distinctive signaling properties. We intend to create synthekines for mouse receptors so that we can study in vivo functional effects, but the reagents needed to create the mouse synthekines are not as readily available. This limitation is explicitly mentioned as a caveat in a paragraph in the Discussion section.

Second, we engineered synthekines using a very limited available inventory of binding modules to the ECD’s of the receptors, which constrained our ability to probe structure-activity space for each synthekine. There are nearly an unlimited number of ways that synthekines could be constructed using antibody domains and other binding scaffolds. There is an almost equally diverse space for how the modules can be linked to each other (e.g. linker peptides of different lengths, zippers, bi-specific Fc). Each of these different types of synthekines will likely result in different levels, and profiles of signaling activity due to alternative orientations and proximities of the receptor ECDs. We refer the reviewers to Moraga et al., 2015 (among other published works cited within this paper) that demonstrated that the signaling amplitude and phosphorylation cascade is keenly sensitive to the receptor dimer distance and geometry imposed by the ligand. In the current study, we made the synthekines in the simplest way using Gly-Ser linkers (and zippers for the cytokine/RTK fusion). We believe they exhibit far less signaling strength that is possible through alternative dimerizing strategies that, for example, will compel the receptors to be closer together. Indeed, we show in Figure 2 that ‘rotating’ the ICD of a receptor through Alanine insertion can convert a non-signaling heterodimer into a signaling dimer when the ECDs are constrained to the geometry of the IL-1 receptor ECD complex. If we did this for each synthekine combination it would likely result in a similar spectrum of activities depending on geometry, but in this initial study we sampled a very limited conformational space for the dimer geometry. Tuning these parameters to maximize signaling is something that will have to be explored in depth in future studies within a given system.

*a) A central question is: do synthekines have unique biological properties or just manifest the combined, yet attenuated, activities of their parent cytokines? The distinction is crucial for understanding mechanism of action. On one hand, synthekines appear to behave like weak agonists; the STAT profile of cells exposed to SY1 mostly resembles that of cells exposed to low doses of IL-4 and IFN, while the STAT profile of cells exposed to SY2 mostly resembles that of cells exposed to low doses of IL-4 and IL-2 (Figure 3—figure supplement 1). Mass cytometry data for other downstream signalling pathways tends to support this view – most analytes are either unchanged or reduced compared to parent cytokines (Figure 4) – and the authors acknowledge such quantitative effects. On the other hand, they also provide evidence for qualitative differences. In particular, the ratio of STATs activated by synthekines appears to be distinct from that of parent cytokines; SY1 elicits a STAT6>STAT5>STAT1>STAT3 pattern instead of the STAT6>STAT1>STAT3>STAT5 pattern seen with IL-4 plus IFN, while SY2 elicits a STAT1>STAT6>STAT5>STAT3 patterns instead of the STAT6>STAT5>STAT3>STAT1 pattern seen with IL-4 plus IL-2. However, better understanding of downstream consequences is necessary to confirm that these changes translate to unique functions.*

In the revised manuscript we have been careful to discuss the signaling outputs elicited by synthekines as ‘distinct,’ and we are cautious about using the term “new” until we have more invasive functional data. Thus we have modified the title and text throughout to express this caveat. We anticipated that each synthekine could elicit some signals resembling those of the parent cytokine for each receptor chain, and at the same time initiating new hybrid signals, resulting in a complex, but importantly *different*, overall signaling outcome. It is not surprising there would be some overlap between the STAT activation profiles of synthekines and their parent cytokines, considering that there are only seven STAT molecules that can be activated. Indeed, many cytokines activate identical STAT molecules but exhibit very different biological outcomes. For instance, IL-2, EPO, TPO GH all activate STAT5; IL-10 and IL-6 both activate STAT3, but the cytokines effect highly divergent functional activities. We believe that a better predictor of biological responses induced by cytokines is the ratio of STAT activation induced. Previous reports have shown that changes in STAT activation ratios dramatically alter the biological responses activated by cytokines (Gli MP et al., 2012; Sharif et al., 2004; Thomas et al., 2011). As the reviewer noted, the STAT ratios engaged are completely different between the genome-encoded cytokines and synthekines. For instance, although the STAT response to SY2 most closely resembles that of cells exposed to low doses of IL-2 and IL-4 (Figure 3 and Figure 3—figure supplement 1), both the maximum levels and rank order of STAT signals are distinct for synthekine versus combination cytokine treatment.

In the revised manuscript, we have included unsupervised clustering analysis (PCA analysis) of the high-throughput signaling data generated with the cytokines and synthekines. This analysis revealed that the signaling programs activated by synthekines and by genome-encoded cytokines clustered separately, indicative of behavior that is unique from that of the parent cytokines. This new graph has now been included as Figure 4—figure supplement 1. We agree that it will be interesting to see how these differences in signaling translate into unique functional activities in future work.

*The authors offer one set of experiments where they compare effects of synthekines and parent cytokines on PBMC cytokine production but the observed differences are subtle and the data are difficult to interpret vis-a-vis what we know about the parent cytokines. For instance, why does SY1 promote IL-17 production when its dominant signaling moieties, STAT6, STAT5 and STAT1, are all thought to inhibit this process (Yang et al., Nature Immunology 2011; Harrington et al., Nature Immunology 2005)? Why does it promote IL-13 production when its ability to activate STAT6, a major driver of this cytokine, is greatly diminished? And, in a related question, why does IL-4 fail to induce IL-13 despite unbridled STAT6 activation? It also bears noting that this experiment strongly supports the idea for SY2 as just a weak agonist; IL-2-driven effects are completely lost, IL-4-driven changes are diminished and there is only one unanticipated analyte, eotaxin, which is known to be regulated by IL-4 in a STAT6-dependent manner (Hoeck et al., JI 2001).*

The question of why certain cytokines are secreted in response to certain STAT activation is difficult to answer. We do not think the correlations described in the literature between transcription factor activation and gene expression are necessarily always predictive. Indeed, synthekines now provide a new tool to perturb cytokine signaling and explore these questions. We believe there is tremendous plasticity in the downstream signaling cascades of kinase-linked receptors that can be perturbed with these extracellular ligands. For example, for SY1 and IL-17 secretion, there are several possible explanations. First, whereas our data show a comprehensive analysis of the phosphorylation patterns of several STATs, the papers cited do not show what happens to other STATs when a given STAT is absent, i.e. whether the amount, phosphorylation or kinetics of activation of other STATs change. These parameters could play a role in altering gene expression. Indeed, previous reports have shown that modification of the levels of a given STAT impact the activation of other STAT molecules (Park et al., Immunity, 2000) meaning that previous findings under different STAT activation conditions are not necessarily incompatible with our results. Second, as the reviewer points out in 1(b), it is possible that unnatural receptor pairs lead to the formation or changes in the amounts of STAT heterodimers, which could account for the unique/unexpected functions observed. Detailed studies of synthekines in mouse models in future studies will reveal more insight into their likely complex physiological effects.

With regard to SY1/IL-4 and IL-13 secretion: We have not been able to find any paper describing a connection between STAT6 activation and IL-13 secretion. IL-13 and IL-4 both activate STAT6, but to the best of our knowledge, IL-13 is not a STAT6-dependent gene. We have found reports showing that STAT5 activation, mainly by IL-2, leads to secretion of IL-13 (Hoshimo et al., 1999), and our data fully agree with these observations.

*b) If we consider that the engineered receptors and synthekines presented in this paper drive juxtaposition of cytokine receptors that are not normally associated with one another, then the issue of STAT heterodimers becomes highly relevant. Certain cytokine receptors are already known to elicit heterodimer formation (Delgoffe & Vignali, JAKSTAT 2013) and it is possible that the unnatural receptor combinations described here could increase/decrease their prevalence or even form previously unseen species, such as STAT1/STAT6 heterodimers downstream of SY1, thus providing a molecular basis for unique functions.*

We agree with the reviewer that this is an important and exciting possibility that probably will greatly influence the immune-modulatory activities of the synthekines. We have now expanded the Discussion to include this possibility.

*c) Flow cytometry based assays may be more suitable for assessing synthekine function. The authors could compare expression of proteins that are known to be regulated by a particular STAT (e.g. LIF for STAT5, GATA-3 for STAT6, CD274 for STAT1, IL-21 for STAT3) in strictly defined PBMC populations (i.e. CD4^+^ T cells, monocytes) that have been treated with synthekines or escalating doses of one or both parent cytokines (as in Figure 3—figure supplement 1). In addition, since each of the parent cytokines is known to impact helper T cell differentiation (i.e. Th1 vs. Th2 vs. Th17), this may also be a useful model for assessing cellular effects of synthekines. Alternatively, defined PBMC populations could be sorted, treated with synthekines and/or parent cytokines and expression of known STAT-regulated genes (e.g. interferon signature genes) measured by PCR or, ideally, by microarray or RNA-seq.*

While we agree that the experiments suggested would provide more insight into specific signaling activities of synthekines, we do not believe that they would lend a great deal more functional insight, short of in vivo models. Measuring synthekine-induced cytokine secretion or gene expression will only tell us whether a synthekine is more or less potent than its parent cytokines in achieving a certain effect, but not whether the synthekine has a different function from the parent cytokines. The predictive power of ex vivo functional assays with human cells is limited.

Nevertheless, we performed a similar experiment to the one proposed by the reviewers, looking at the response of isolated dendritic cells to synthekines versus natural cytokines (Figure 9). We wanted to take advantage of the phenotypic plasticity of dendritic cells, which can exhibit different functions depending on the cytokines environment, to assay whether synthekines elicit new bioactivities. To that end, we purified monocytes from PBMCs and stimulated them with GM-CSF alone or supplemented with Super-2, IL-4, SY1 or SY2. IFN stimulation led to monocyte cell death so we did not use it in this set up. On day seven we assayed the cultured cells for known markers of DCs differentiation. As shown in Figure 9, in most instances the synthekines had a mild effect, in all cases, weaker than the effect elicited by IL-4, the cytokine for which this experimental set up was optimized. However, this does not mean that SY1 and SY2 are just weak agonist versions of IL-2 and IL-4, they may have entirely different functional activities from the ones assayed here. For example, we previously created a partial agonist version of IL-2 (Mitra et al., Immunity 2015) that in vitro, simply showed that it only induced 50% Emax of STAT5 activation, yet it showed cell type activation biases on T cells, and quite distinct in vivo activities (unpublished). These results highlight again that ex vivo assays are not sufficient to predict the functional properties of synthekines.

Author response image 1.CD14^+^ monocytes were isolated (>97% purity) from PBMCs obtained from healthy blood donors by density centrifugation and CD14 magnetic microbeads separation.O.5-1 x 10^6^ CD14^+^ monocytes were cultured with GM-CSF alone or with the indicated cytokines and synthekines for seven days. Fresh ligands were added on days 2 and 4. On day 7 cells were processed and stained for the indicated dendritic cell surface markers.**DOI:**
http://dx.doi.org/10.7554/eLife.22882.017

*d) Another possibility is whether synthekines can antagonize their parent cytokines in terms of signaling and/or downstream effects. Thus, experiments where synthekines are added to cells together with one or both parents would be informative.*

This is a very interesting question and there are many complex scenarios for how this could happen. If one or both of the receptor binding module(s) in the synthekine are antagonistic to the parent cytokine of each receptor in the new dimer (i.e. competitive for binding to the receptor ECD), then it would antagonize the parent cytokine in vivo and this could add an additional layer of complexity to the functional effects of the synthekines. Alternatively, the synthekine signal could potentiate or dampen the signal of the parent cytokine through intracellular cross-talk, which is entirely possible. In this respect, the synthekines could have both agonistic and/or antagonistic effects to the other cytokines. Additionally, one could make synthekines that allow the parent cytokine to simultaneously engage the receptor (essentially giving the receptor a second signaling ligand), so the activity of the synthekine would be layered on top of the parent cytokine. Such dual possibilities will certainly impact the in vivo functions of the cytokines, however it would be very difficult to parse out which effects are due to the relative actions of parent versus synthekine, and antagonism versus agonism.

*2) There is also a lack of cohesiveness between the three principal components of the manuscript – the first two components (chimeric receptors and synthekines) clearly inform one another but the third section (about cytokine-RTK fusions) seems tangential. The latter constructs are certainly interesting from a basic science perspective – in fact, the data presented illustrate that Jaks can phosphorylate RTKs (at least EGFR and KIT) but not vice versa – and exciting in terms of therapeutic potential, but some attempt at improved integration of the manuscript would be desirable.*

We agree that, as currently presented, our finding on a synthekine that dimerizes a JAK/STAT receptor (TPO) with an RTK (cKit) is a bit disjointed with the main thrust of the paper on JAK/STAT cytokine receptors. However, this is a very important experiment and shows that the combinatorial scope of synthekines is not just limited to kinase-linked receptors of the same class. We have modified the Abstract and Introduction to better integrate the JAK/STAT cytokine and RTK sections. We have rewritten sections of the results to better explain these experiments, our interpretations of the results and the apparent asymmetry of the trans-phosphorylation, as the reviewer notes. We suspect that some explanation to this observation lies in the particular geometry that the SY5 synthekine dimerizes cKIT with TpoR, and that other synthekines may or may not elicit such asymmetry. We also included citation of several papers that have reported observing cross-talk between RTK and JAK/STAT receptors on natural cells, which lends support to our data presented here.